# Synergistic Upregulation of Extracellular Vesicles and Cell-Free Nucleic Acids by Chloroquine and Temozolomide in Glioma Cell Cultures

**DOI:** 10.3390/ijms26199692

**Published:** 2025-10-04

**Authors:** Aleksander Emilov Aleksandrov, Banko Ivaylov Bankov, Vera Lyubchova Djeliova, Georgi Georgiev Antov, Svetozar Stoichev, Roumyana Silvieva Mironova, Dimitar Borisov Iliev

**Affiliations:** 1Roumen Tsanev Institute of Molecular Biology, Bulgarian Academy of Sciences, 1113 Sofia, Bulgaria; aleksander_aleksandrov@proton.me (A.E.A.);; 2Laboratory of Genome Dynamics and Stability, Institute of Plant Physiology and Genetics, Bulgarian Academy of Sciences, 1113 Sofia, Bulgaria; antov8107@abv.bg; 3Institute of Biophysics and Biomedical Engineering, Bulgarian Academy of Sciences, 1113 Sofia, Bulgaria

**Keywords:** extracellular vesicles, exosomes, autophagy, temozolomide, chloroquine, DNA damage response

## Abstract

Extracellular vesicles (EVs) secreted by glioblastoma multiforme and other types of cancer cells are key factors contributing to the aggressiveness of the disease and its resistance to therapy. Chloroquine (CHQ), a lysosomal inhibitor, has shown potential as an enhancer of temozolomide (TMZ) cytotoxicity against glioblastoma cells. Since both CHQ and TMZ are known to modulate EV secretion, we sought to investigate their potential interplay in this process. Simultaneous treatment of TMZ-sensitive (U87-MG) and TMZ-resistant (U138-MG) glioblastoma cells with TMZ and CHQ led to a synergistic upregulation of EV secretion. Although CHQ did not enhance the TMZ cytotoxicity in U87-MG cells, it synergized with the latter to upregulate the release of extracellular nucleic acids implicating activation of unconventional secretory pathways. Synergistic upregulation of the autophagy markers LC3B-II and p62 by CHQ and TMZ in both cells and EVs indicates that secretory autophagy is likely involved in the observed unconventional secretion. Moreover, a significant enrichment of caveolin-1 in small EVs highlights their potential role in modulating tumor aggressiveness. The synergy in EV upregulation was not confined to the specific biological activity of TMZ and CHQ; similar effects were observed upon co-treatments with CHQ and etoposide (a topoisomerase inhibitor) and TMZ and Bafilomycin A1 (another lysosomal inhibitor). Heightened EV release was also observed in THP-1 monocytes and macrophages treated with Bafilomycin and TMZ, highlighting a broader, cell-type-independent mechanism. These findings indicate that combined DNA damage and lysosomal inhibition synergistically stimulate secretory autophagy and EV release, potentially impacting the tumor microenvironment and driving disease progression.

## 1. Introduction

Glioblastoma multiforme (GBM) is the most common and aggressive type of brain cancer [1]. Currently, the therapy for GBM includes surgical resection, radiotherapy and chemotherapy with temozolomide (TMZ), a DNA alkylating agent, which does not prevent the inevitable recurrence of the tumor at or around the primary site [2]. GBM tumors are characterized by a highly heterogeneous microenvironment containing both cancerous and different types of non-cancerous cells [3]. The intra-tumoral GBM cells are engaged in complex interactions that delineate the aggressiveness of the disease, and in which extracellular vesicles (EVs) are essential factors [4].

EVs are membrane-bound particles that do not possess a functional nucleus [5]. They are found in bodily fluids and in conditioned cell culture medium, and are vastly heterogeneous, due to their differing intracellular origins and mechanisms of biogenesis, which reflect on the diversity of their size, composition and functional properties [6]. EVs function as intercellular vehicles for bioactive molecules including lipids, proteins and nucleic acids, allowing effective communication between different types of cells [7]. EVs are abundantly produced by cancer cells and are implicated in the etiology and development of malignancy by promoting angiogenesis, invasion, evasion of apoptosis and resistance to chemotherapy, including TMZ [8,9].

Autophagy is a catabolic process whose canonical function is degradation of intracellular components including, in the case of macroautophagy, whole organelles. It is induced upon stress and starvation, and its main purpose is to supply cells with nutrients, while eliminating potentially cytotoxic intracellular waste such as dysfunctional organelles, protein aggregates and damaged DNA [10]. Autophagy-related processes have also been found to be involved in unconventional secretory pathways, including biogenesis and secretion of EVs [11,12].

Studies have implicated autophagy in the etiology of GBM and the development of resistance to treatment [13,14,15]. More specifically, upon TMZ-induced cell cycle arrest, autophagy improves cell viability by removing damaged organelles and proteins and, consequently, reducing cellular stress, which contributes to development of TMZ resistance [16,17]. Reportedly, chloroquine (CHQ), an inhibitor of late-stage autophagy and an inducer of EV secretion, potentiates radiosensitivity and enhances TMZ cytotoxicity in malignant gliomas through autophagy-dependent and independent mechanisms [12,18,19,20,21,22]. Furthermore, studies have implicated caspase activity as well as phosphatidylinositol 3-kinase (PI3K)/Akt and p53-dependent signaling pathways in the CHQ-enhanced TMZ cytotoxicity [15,20,23].

In clinical trials, CHQ has shown potential benefit as an adjuvant for therapy-refractory cancers including GBM [24,25]. TMZ is also known to modulate EV release [26,27]; however, to our knowledge, at the time of writing, there are no reports on the possible interaction between TMZ and autophagy inhibitors regarding regulation of EV secretion.

In the current study, we have used in vitro GBM models (U87-MG and U138-MG cells) to demonstrate that TMZ synergizes with lysosomal/late-stage autophagy inhibitors (CHQ and Bafilomycin A1) to induce secretion of EVs and extracellular nucleic acids.

## 2. Results

### 2.1. CHQ Significantly Enhances the Cytotoxic Effect of TMZ on TMZ-Resistant (U138-MG) but Not TMZ-Sensitive Glioma Cells (U87-MG)

As apoptotic bodies may contribute significantly to the pool of EVs, an important aspect in analyzing the rates of active EV secretion is to investigate the rates of apoptosis under specific experimental conditions. In this study, we used two GBM cell lines to test the potential interplay between TMZ and CHQ in EV secretion. U87-MG cells are known to be TMZ-sensitive as they do not express the O6-Methylguanine-DNA Methyltransferase (MGMT). In contrast, U138-MG cells do express this enzyme and have been demonstrated to be more resistant to TMZ compared to cells that do not express MGMT [28]. Typically, in in vitro studies of EVs, serum-free medium or medium with low (2–5%) concentrations of EV-depleted FBS are used to avoid contamination of samples with serum EVs, and to reduce the amounts of albumin and other lipid-binding proteins which associate with EVs. In the current study, we compared the effect of 10 µM CHQ on the cytotoxicity of 400 µM TMZ in U87 and U138 cultures after 48 h of incubation in medium with 5% EV-depleted FBS. These concentrations were selected based on prior reports: TMZ IC_50_ values for U87 cells are typically ~100–300 µM (24–72 h), whereas U138 cells are considerably more resistant, with IC_50_ values approaching 1 mM. CHQ is generally cytotoxic for glioma cells at concentrations > 10 µM, but is widely used at 10 µM in studies examining its role as an autophagy inhibitor and sensitizer to TMZ. A detailed summary of reported IC_50_ values is provided in Appendix A. In agreement with previous reports [28], the PI/Annexin V-FITC staining demonstrated that the U87 cells were more sensitive to TMZ compared to the U138 cells—10.6% vs. 5.92% late apoptotic/necrotic cells, respectively (Figure 1A,B). The addition of CHQ significantly increased the percentage of TMZ-induced PI-positive late apoptotic/necrotic cells in U138 (9.76%), but not in U87 cultures (10.63%). Concurrently, combined treatment with TMZ + CHQ significantly decreased the percentage of live U138 cells (87.25%), when compared to treatment with TMZ alone (91.56%), which was not observed in U87 cultures (87.42% vs. 88.23%, respectively).

### 2.2. TMZ and CHQ Synergistically Induce Secretion EVs Containing Autophagy Markers and Caveolin-1 in Cultures of U87 MG Cells

Protein samples of large EVs (LEVs) and small EVs (SEVs) from DMSO-treated controls, and cells treated with 400 µM TMZ and 10 µM CHQ alone and in combination, were compared using WB. The autophagy markers (LC3B-II and p62) and the exosome markers (alix and syntenin-1) were synergistically upregulated by the combination of TMZ + CHQ in LEVs and SEVs, respectively (Figure 2A,B). CD63, another widely used marker for EVs was detectable in all samples, including the DMSO-treated controls; however, it also showed significant upregulation in both LEVs and SEVs upon treatment with TMZ + CHQ. Since CD63 is heavily glycosylated and typically forms smears on WB, for improved accuracy, it was quantified with dot blots. Similarly to CD63, caveolin-1 was detectable in all of the LEV samples, including the controls. However, in contrast to CD63, it was not present in SEVs from control samples, and, like the exosome markers, it was synergistically upregulated by TMZ and CHQ in SEVs. The mitochondrial marker—cytochrome C1 (CYC1)—which is not expected to be present in sufficiently purified EVs was not detected in any of the samples, indicating adequate purity of the EVs obtained in the current study. The hydrodynamic size range of the EVs was determined with dynamic light scattering (DLC) (Figure 2C). Only samples from cells treated with both TMZ and CHQ yielded reliable measurements, most likely due to the very low number of EVs in the other samples. The size of the TMZ + CHQ EVs ranged between 50 and 150 nm for SEVs and 150–500 nm for LEVs, which is typical for exosomes and microvesicles, respectively.

In whole cell lysates (WCL), TMZ alone did not induce significant accumulation of p62 or the lipidated, membrane-bound LC3B-II; however, these autophagy markers were significantly upregulated in the TMZ + CHQ samples compared to individual treatments and vehicle controls. (Figure 2D).

The U138-MG cells treated with TMZ and CHQ responded similarly to U87 cells—by robust synergistic upregulation of LC3B-II and p62 in LEVs and caveolin in SEVs (Figure 3A,B). Compared to LEVs, the levels of p62 were much lower in SEVs, but it was also synergistically upregulated in the latter. Although, as mentioned above, in U138 cultures the TMZ + CHQ treatment had a significantly higher cytotoxic effect, as compared to TMZ alone, CYC-1 was not detectable in any of the EV samples. This indicates that the observed effect is due to active secretory mechanisms, rather than passive release of material from dying cells. As in U87, LC3B-II and p62 were significantly upregulated in WCLs of U138 cells following treatment with TMZ + CHQ, compared to controls, as well as individual treatments. (Figure 3C,D).

### 2.3. CHQ Induces Accumulation of LC3B in the Lumen of CD63-Positive Amphisomes/Autolysosomes; LC3B and Caveolin-1 Colocalize Within the CHQ-Induced LC3B Aggregates

Inside cells, caveolin-1 is mostly associated with caveolae–membrane invaginations, whose size is similar to that of exosomes (50–100 nm). Although this protein is known to be secreted by cancer cells in large, plasma membrane-derived oncosomes with a prognostic value for metastatic cancer [29], caveolae are not known to be directly released from the plasma membrane. This raises a question about the source of caveolin-1 in the TMZ + CHQ-induced SEVs. Here we demonstrate that CHQ treatment effectively leads to accumulation of large LC3B aggregates in CD63-positive intracellular vacuoles, indicating that these are autolysosomes and/or amphisomes (Figure 4A). The lysosome inhibitor also induced colocalization between LC3B and caveolin-1 within these relatively large intracellular vesicles (Figure 4B), which might represent the source of the TMZ + CHQ-induced EVs, including the caveolin-1-laden SEVs.

### 2.4. CHQ and Etoposide Synergistically Upregulate LC3B-II and p62 in LEVs

In order to find out whether CHQ may synergize with other genotoxic agents to upregulate EV secretion by glioma cells, we performed experiments with etoposide—a chemotherapeutic agent which induces DNA damage through topoisomerase II inhibition [30]. We used 20 µM etoposide after carefully titrating its concentration to achieve DDR levels (by measuring γH2A.X) comparable to those induced by 400 µM TMZ. The results shown in Figure 5A,B demonstrate that, similarly to TMZ, etoposide significantly enhances the CHQ-induced secretion of the autophagy markers by U87 cells.

### 2.5. CHQ Enhances the TMZ-Induced DDR as Indicated by γH2AX Upregulation but It Does Not Affect the Accumulation of Damaged DNA or the TMZ-Induced G2/M Cell Cycle Arrest of U87 Cells

CHQ is known to induce DNA damage and G0/1 cell cycle arrest in certain cell lines [31]. In the current study, we did not observe any significant upregulation γH2AX by CHQ alone in U87 cells, probably due to its relatively low concentration. In fact, CHQ alone led to a slight but consistent, although not statistically significant, downregulation of γH2AX, which contrasted with the synergistic upregulation following TMZ + CHQ treatment observed with both WB (Figure 6A,B) and immunocytochemistry (Figure 6C,D).

The direct effect of CHQ and TMZ on DNA integrity was investigated with an alkaline comet assay which detects both single- and double-strand DNA breaks (Figure 6E). The results demonstrate that TMZ treatment induces DNA damage, as indicated by the significantly increased amount of DNA in comet tails. CHQ, alone or in combination with TMZ, did not increase the extent of DNA damage, nor did it significantly influence the TMZ-induced G2/M cell cycle arrest. (Figure 6F).

### 2.6. CHQ and TMZ Synergistically Induce Secretion of Cell-Free Nucleic Acids in U87 Cultures

The concentration of cell-free DNA was directly measured in conditioned supernatants of cells grown in FluoroBrite™ DMEM (Gibco, Cambridge, MA, USA) using a Qubit™ 1X dsDNA High Sensitivity (HS) assay kit (Invitrogen, Carlsbad, CA, USA) (Figure 7A). As compared to DMSO controls or individual treatments, significant upregulation of cell-free dsDNA was observed in samples treated with TMZ + CHQ.

To examine the size distribution of nucleic acids (NAs) in conditioned medium, extracellular NAs were isolated from SNs (precleared with 500× *g* and 1500× *g* centrifugations) with phenol/chloroform/isoamyl alcohol extraction, without DNase or RNAse treatment. Total extracellular/cell-free NA samples, isolated from equal volumes of conditioned SN from cells treated for 48 h, were run on an ethidium bromide-stained agarose gel (Figure 7B). The samples contained NA molecules, ranging in size from ~200 to ~20,000 bp, referred to as the DNA ladder. Treatment of the NA samples with nucleases prior to analyzing them on an agarose gel showed that while the smear > 500 bp was, by and large, composed of DNA, most of the molecules < 500 bp were sensitive to RNase A (Figure 7C). The A260 absorbance of the samples analyzed on the agarose gel showed substantial increase in the TMZ + CHQ sample, compared to the control and the single treatments. Note that due to the differing extinction coefficients of dsDNA, RNA and ssDNA, accurate and absolute quantification of mixed NA samples cannot be performed using spectrometry.

As previously reported, the active secretion of cell-free DNA may be either EV-dependent, or amphisome-dependent/exosome-independent [11]. To investigate the association of TMZ + CHQ-induced cell-free NA with EVs and particles (EVPs), we analyzed PI-stained EV samples with flow cytometry. Since flow cytometry of small EVs, such as exosomes, is hampered by background noise and swarming effects, we included only samples of large EVs in the analysis. Propidium Iodide (100 ng/mL) was added to LEV samples resuspended in PBS, and PI+ EVPs were quantified, using low flow rate and fixed time (50 s). The results show that TMZ alone causes a significant increase in the number of PI + EVPs (Figure 7D), which is further significantly enhanced by the addition of CHQ. (Figure 7E) Treatment of the EVPs with 1% Triton X100 lowered the number of PI+ EVPs in all of the treatments by approximately 50%, indicating that the samples contain both EVs and Triton-resistant particles associated with NA.

### 2.7. Response of THP-1 Monocytes to TMZ and Autophagy Inhibitors; TMZ Synergizes with BafA1, but Not with CHQ, to Induce EV Secretion

Next, we wanted to find out whether the synergistic induction of EVs by CHQ and TMZ may be observed with other cell types. Monocytes are abundant in GBM tumors, comprising up to 30–50% of the tumor cells [32], and they are hypersensitive to the cytotoxic effect of TMZ [33], which bears implications for GBM therapy. In the current study, we used cultures of THP-1 monocytes to investigate their response to TMZ and CHQ. Under conditions identical to those used with GBM cells, there was no detectable upregulation of LC3B-II, p62 or the exosomal marker syntenin-1 in EV samples from THP-1 cells treated with TMZ and CHQ (Figure 8A,B). Therefore, in these experiments, we included Bafilomycin A1 (BafA1) which, like CHQ, is often used to inhibit lysosomal function and, consequently, degradative autophagy. Treatment of the THP-1 cells with 50 nM BafA1 alone induced secretion of the tested markers, which was further significantly enhanced by TMZ (Figure 8A,B). The DLS analysis confirmed that the TMZ + BafA1 treatment had induced secretion of vesicles with typical size distribution profiles for exosomes and LEVs, with modes of 126 and 239 nm, respectively (Figure 8C). In WCLs of THP-1 cells, both CHQ and BafA1 induced accumulation of LC3B-II; however, only BafA1 upregulated p62 (Figure 8D). Unlike in GBM cells, TMZ, whether alone or in combination with the lysosomal inhibitors, did not affect LC3B-II and p62 levels in monocytes.

The results from the cell viability assay confirmed that, compared to GBM cells, the THP-1 monocytes were more sensitive to the TMZ treatment, with ~40% PI-positive cells following 48 h treatment with 400 µM TMZ (Figure 8E). While CHQ treatment did not affect THP-1 viability, neither alone nor in combination with TMZ, BafA1 was cytotoxic by itself, and significantly more so in combination with TMZ.

Experiments with higher CHQ concentrations in THP-1 monocytes (Appendix A) and PMA-induced THP1 macrophages (Appendix A), showed that 50 and 100 µM CHQ induce LC3B-II, p62 and syntenin-1 release. However, the LEV samples contained a high amount of CYC1—an indicator for contamination of the EV samples with cell debris, which reflects the significant cytotoxic activity of high doses of CHQ (Appendix A). Unlike BafA1, neither 10 nor 50 µM CHQ could inhibit the proteolysis of endocytosed DQ-OVA substrate (Appendix A)—an indication of retained lysosomal activity in THP-1 cells.

### 2.8. TMZ Enhances the Bafilomycin A1 (BafA1)-Induced EV Secretion by U87 Cells

Treatment of U87 with 50 nM BafA1-induced secretion of LEVs, containing LC3B-II, p62, CD63 and caveolin-1, which was significantly amplified by TMZ. (Figure 9A,B) As in THP-1, BafA1 had a significant cytotoxic effect on glioma cells, and with the addition of TMZ, the PI-positive cells exceeded 50%. Brefeldin A (BFA) is an inhibitor of intracellular vesicle transport and EV secretion, an autophagy modulator and an apoptosis inducer [34,35,36,37]; however, while exhibiting cytotoxicity comparable to that of BafA1 (Figure 9C), Brefeldin A did not detectably increase EV secretion in U87 (Figure 9A) or U138 (Appendix A) cell cultures.

## 3. Discussion

This study highlights the capacity of CHQ and TMZ to synergistically activate unconventional secretory pathways, as demonstrated by the enhanced secretion of EVs as well as cell-free DNA and RNA in glioma cell cultures. The additional experiments with etoposide and Bafilomycin A1 suggest that the synergy in EV upregulation is not confined to the specific biological activity of CHQ and TMZ, and is rather very likely driven by DNA damage in the context of lysosomal/late-stage autophagy inhibition.

It has been reported that autophagy is implicated in the secretion of EVPs [38,39,40]. It has also been proposed that impaired autophagosome maturation leads to Rab27a-dependent secretion of autophagosome-derived material, along with classical, multivesicular body-derived exosomes [12]. Unlike the results presented here, in the aforementioned report there was a significant upregulation of EV secretion by CHQ. This could probably be explained by the use of a higher CHQ concentration (25 µM), compared to the 10 µM CHQ used in our study. Above a certain concentration (>10 µg/mL), CHQ may induce apoptosis in glioma cell cultures [23]. Reported IC_50_ values for TMZ and CHQ vary widely depending on assay, serum conditions and exposure time. For U87-MG, TMZ IC_50_ values are generally in the range of ~100–300 µM after 24–72 h, whereas U138-MG cells are markedly more resistant, with published IC_50_ values close to 1 mM under some conditions [28]. In glioma models, CHQ typically exerts cytotoxic effects at concentrations > 10 µM, although it is frequently used at 10 µM in mechanistic studies as an autophagy inhibitor. Our treatment conditions (400 µM TMZ, 10 µM CHQ) therefore represent a supra-IC_50_ dose of TMZ for the sensitive U87 line, but a sub-IC_50_ dose for the more resistant U138 line, consistent with our observations. The chosen CHQ concentration is within the range commonly employed to inhibit autophagy. A summary of reported IC_50_ values is provided in Appendix A.

High levels of apoptosis may potentially obscure active secretion of EVs, due to discharge of apoptotic bodies with a size range overlapping that of classical microvesicles and exosomes [41]. In U87 cells, the synergistic induction of EV secretion by TMZ and CHQ is unlikely to be apoptosis-dependent, since under these conditions CHQ did not enhance the TMZ-induced cytotoxicity. Compared to U87, the U138 cells are more resistant to TMZ (Figure 1B); however, in the latter, CHQ did enhance the cytotoxicity of TMZ. It is unlikely that this could have significantly affected the observed induction of LC3B-II, p62 and caveolin-1 secretion, since cytochrome C1, a marker for contamination with cell debris, was not detected in EVs from U138 cells (Figure 3A).

An interesting observation in the current study, is the synergistic induction of caveolin-1 by TMZ and CHQ in small EVs (Figure 2 and Figure 3). In late-stage cancer, including GBM, the expression of caveolin-1 correlates with the tumor grade, and the protein contributes to the aggressiveness of the disease, as well as its resistance to therapy [42]. Caveolin-1 is a plasma membrane protein, and it is present in LEVs, including plasma membrane-derived oncosomes [43]. It is also enriched in serum exosomes of GBM patients, compared to healthy controls [44]. A proposed mechanism implicates secretory autophagy in the biogenesis of caveolin-1-containing EVs [45], and the data presented here concurs with this model—as inferred by the CHQ-dependent secretion of caveolin-1-loaded SEVs, and the CHQ-induced accumulation of the protein in large amphisomes/autolysosomes (Figure 4).

In addition to acting as a lysosomal inhibitor, CHQ has a DNA intercalating activity, leading to an alteration of the DNA structure and, consequently, ATM activation in the absence of DNA damage [46]. It has also been demonstrated that, at the concentration used in the current study (10 µM), CHQ enhances DNA repair, rather than damage [47]. This may help explain the ostensible discrepancy observed here—enhanced γH2Ax upregulation when CHQ is added to TMZ treatment, without further accumulation of DNA breaks or increased number of cells arrested in G2/M phase (Figure 6). It is also possible that minor increases in damaged DNA within the TMZ + CHQ-treated cells might not be detectable with the comet assay used in the current study. However, fragmented extracellular DNA, as well as RNA molecules, were detectable in conditioned supernatants, and the TMZ-induced EVP-associated extracellular nucleic acids were further synergistically upregulated by the addition of CHQ (Figure 7). In this regard, the work presented here adds to a growing number of studies that have demonstrated relationships between DDR and EV secretion. Namely, DNA damage may influence secretion and composition of EVs and vice versa—EVs may affect cells through regulation of DDR [48]. Damaged nuclear DNA is actively transferred to the cytoplasm, wherefrom it may be expelled out of the cell through both autophagy-dependent [49] and independent [50] unconventional secretory mechanisms. Moreover, EVs, including exosomes, are known to serve as vehicles for export of damaged DNA as well as RNA–DNA hybrids [51,52]. Accumulation of DNA and double-stranded RNA within the cytoplasm may activate cytotoxic, proinflammatory signaling, downstream of cGAS-STING and RIG-I/MDA, respectively, leading to cellular senescence [53,54]. TMZ-induced DDR has been linked to activation of cytoprotective autophagy [55]. In addition to alleviating the harmful effects of the cytoplasmic DNA-triggered signaling pathways, secretory autophagy and EVs are known to contribute to cellular resistance to genotoxic drugs, including TMZ, by enhancing the efflux of the latter [56]. In the context of lysosomal inhibition, a shift towards reliance on secretory autophagy for the elimination of damaged NAs could explain the synergy observed in the current study.

Although we cannot ascertain whether the synergistic EV response to combined treatment with TMZ and CHQ is confined to glioma cells, the results presented here suggest a dependency on cell type. The effect was not observed in THP-1 monocytes and macrophages treated with the same concentrations of CHQ and TMZ as GBM cells, despite the TMZ treatment being more toxic for THP-1, compared to GBM cells. CHQ did upregulate LC3B-II in THP-1 cells, however, p62 expression was affected neither by CHQ nor TMZ (Figure 8D). This suggests that, unlike in GBM cells, the CHQ-dependent inhibition of lysosomal activity in THP-1 cells was relatively inefficient. Higher concentrations of CHQ did induce discharge of EVs and autophagy markers by both THP-1 monocytes and PMA-differentiated macrophages; however, this correlated with the cytotoxicity of the treatment and resulted in contamination of the LEV samples with CYC1 (Appendix A). In this regard, it has been observed that CHQ inhibits autophagy by impairing autophagosome fusion with lysosomes, rather than by affecting the degradative activity of this organelle [57]. In contrast to CHQ, BafA1 upregulated LC3B-II and p62 as well as EV release in both THP-1 (Figure 9) and GBM cells (Figure 9, Appendix A), and TMZ further upregulated the BafA1-induced EV secretion. Also, unlike CHQ, BafA1 did inhibit the lysosomal proteolysis of DQ-OVA substrate by THP-1 monocytes (Appendix A). This highlights the mechanistic differences between BafA1 and CHQ in lysosomal inhibition, and may account for the variability in EV induction observed in THP-1 cultures, treated with the two inhibitors. In both THP-1 and glioma cells, BafA1 exhibited significant cytotoxicity by itself, and even more so, in combination with TMZ, which might have contributed to the robust EVP release. Nevertheless, this does not seem to be the sole reason for the synergy as, in THP-1 cultures, TMZ had a significantly higher cytotoxicity compared to BafA1, without inducing any detectable EV release. In addition, Brefeldin A, which, like BafA1, is an apoptosis inducer [34] and an autophagy modulator [37], but an inhibitor of intracellular vesicle transport and EV secretion [36], did not upregulate EV secretion by GBM cells (Figure 9 and Appendix A). Overall, these data indicate that the synergistic EV upregulation observed in this study is not merely due to increased shedding of apoptotic bodies and cell debris, but is rather dependent on active unconventional secretory mechanisms.

## 4. Materials and Methods

### 4.1. Cell Cultures

The GBM cell lines (U87-MG and U138-MG) and the THP-1 monocytes were obtained from CLS Cell Lines Service GmbH (Heidelberg, Germany). U87-MG and U138-MG cells were maintained in DMEM, high glucose medium (Gibco, Waltham, WA, USA, 41966052) and the THP-1 monocytes were cultured in RPMI 1640 Medium, HEPES (Gibco, 52400041) supplemented with 10% fetal bovine serum (Gibco, 10270106), 100 units/mL of penicillin, 100 µg/mL of streptomycin (Gibco, 15140122) and 0.25 µg/mL of Amphotericin B (Gibco, 15290026) at 37 °C in 5% CO_2_.

### 4.2. Reagents and Primary Antibodies

Bafilomycin A1 (SML1661), TMZ (PHR1437), 1 mM solution of Ionomycin in DMSO (I3909) and Phorbol-12-myristate-13-acetate (524400) were obtained from Merk. CHQ diphosphate salt (455240250) and Brefeldin A (00-4506-51) were purchased from Thermo Scientific Chemicals (Waltham, MA, USA).

Primary antibodies were obtained as follows: from Cell Signaling Technology (Danvers, MA, USA): mouse anti-LC3B (83506S) and rabbit anti-caveolin-1 (3267S); from Abcam (Waltham, MA, USA): mouse anti-CD63 (ab59479) and mouse anti-alix (ab117600); from Merck (Darmstadt, Germany): rabbit anti-actin (a5060); from Santa Cruz Biotechnology (Dallas, TX, USA): anti-γH2AX (SC-517348); from Novus Biologicals (Centennial, CO, USA): rabbit anti-LC3B (NB100), anti-cytochrome c1 (NBP1-86872), anti-p62 (NBP1-49955) and anti-syntenin 1 (NBP1-86979).

### 4.3. Apoptosis and Cell Viability Assay

Cells were treated in 24-well plates (2 × 10^5^ cells/well) in 500 µL of DMEM with 10% normal FBS, 5% exosome-depleted FBS or without FBS for 48 h. The cells were washed with PBS, detached with 100 µL of 1X trypsin (Thermo Fisher, 15400054), washed again with PBS, 5% FBS to neutralize the trypsin and stained with Annexin V-FITC/PI Apoptosis Detection Kit from Thermo Fisher (BMS500FI) as per the manufacturer’s protocol. The samples were analyzed with BD FACSCalibur™ Flow Cytometer and the data was processed with the Cyflogic software (version 1.2.1) (https://www.cyflogic.com).

### 4.4. Isolation of EVs

Cells were seeded in 6-well plates (7.5 × 10^5^/well) in standard culture medium and left to adhere overnight. The adherent cells were washed with serum-free medium and treated for 48 h in DMEM with 5% exosome-depleted fetal bovine serum (Gibco, A2720803)—1.5 mL/well. Samples (conditioned medium from 2 wells/sample) were processed with sequential centrifugations as follows: 500× *g* for 5 min, 1500× *g* for 15 min and 10,000× *g* for 40 min. The 10,000 g pellets containing large EVs were lysed with 2x Laemmli sample buffer (35 µL/sample). The post-10,000× *g* supernatant (SN) was processed for isolation of small EVs using the PEG precipitation protocol [58]. Briefly, equal volumes of 16% Polyethylene Glycol 6000 (Merck, 8074911000) prepared in 1M NaCl solution were mixed with the SNs and the samples were left at 4 °C overnight (at least 12 h). The samples were centrifuged in a tabletop centrifuge at 3500× *g* for 1 h at 4 °C. The supernatant was aspirated and the samples were subjected to a brief centrifugation (2 min at 4000× *g*) to allow for complete removal of the PEG solution. The resulting pellets were dissolved in 35 µL of 2x Laemmli sample buffer (1 μL per 4 × 10^4^ secreting cells) and stored at −20 °C until further use.

### 4.5. Immunoblotting (Western and Dot Blotting)

Protein samples from whole cell lysates (WCL) and EVs released from equal numbers of cells (4 × 10^5^) were heated at 98 °C for 10 min in sample buffer supplemented with 50 mM dithiothreitol, separated using sodium dodecyl sulfate-polyacrylamide gel electrophoresis (SDS-PAGE; 15% acrylamide) and transferred onto Immobilon-P Membrane, PVDF, 0.45 µm (Thermo Fisher, Waltham, MA, USA, 88518). Western blots were blocked in 5% bovine serum albumin (Sigma Aldrich, Darmstadt, Germany, A9647) in Tris-buffered saline with TX100 (TBST; pH 7.4, 10 mM Tris-HCl, 140 mM NaCl, 0.5% v:v Triton X-100) for 1 h and incubated with primary antibodies overnight at 4 °C. Next, the blots were washed 3x with TBST and incubated with secondary HRP-conjugated anti-mouse IgG (Thermo Fisher, 31430) and anti-rabbit IgG (Thermo Fisher, 31460) antibodies (1:20,000 dilution) at room temperature for 1 h. After extensive washing with TBST, the blots were developed with SuperSignal™ West Pico PLUS (Thermo Fisher, 34580) or SuperSignal™ West Femto Maximum Sensitivity Substrate (Thermo Fisher, 34095), depending on the strength of the signal. Imaging of the blots was performed on a C-DiGit Blot Scanner (LI-COR Biosciences—GmbH, Bad Homburg, Germany) and the densitometry was performed with the Image Studio™, vesion 6.0 software.

For the CD63 dot blotting, non-reduced EV samples in Laemmli buffer were spotted in duplicates of 1 µL onto Protran^®^ Supported nitrocellulose membrane (GE10600020), which was then processed as described for the Western blot membranes.

### 4.6. Dynamic Light Scattering

EVs, isolated as described above, were resuspended in 1x PBS and analyzed with Zetasizer Nano ZS analyzer (Malvern Instruments, Malvern, UK). Type of measurement: dynamic light scattering (DLS); analysis model: multiple narrow modes. The cuvette cell for size distribution measurements was DTS0012 plastic disposable 10 × 10 cell. Three replicate measurements were taken for each sample and the data for each size point was presented as percent of the total particle number.

### 4.7. Fluorescence Imaging

Cells (2 × 10^5^ cells/well) were seeded in 24-well plates on 13 mm round cover glasses, #1,5 (VWR, MARI0117530) and left to adhere overnight. The cells were treated in DMEM with 5% exosome-depleted FBS for 48 h, fixed/permeabilized with ice-cold methanol for 10 min and blocked with 5% BSA in PBS for 30 min. Primary antibodies (anti-LC3B—1:400, anti-CD63—1:200, anti-γH2AX—1:200 and anti-BrdU—1:50), were added for 1 h, followed by secondary anti-rabbit IgG Alexa Fluor 555 (Cell Signaling, 4413S), anti-mouse IgG Alexa Fluor 488 (Cell Signaling, 4408S) and anti-mouse AlexaFluor 647 (Invitrogen, A-21237) antibodies for 45 min at 1:1000 dilution. After washing with PBS and staining with DAPI, the coverslips were mounted on slides using ProLong™ Diamond Antifade Mountant (Thermo Fisher, P36965). The cells were imaged on a Leica Stellaris 8 confocal system with 100x oil-immersion objective. Images were processed and analyzed with the ImageJ software (version 1.54q).

### 4.8. Analysis of Extracellular NA

Conditioned supernatants were precleared with sequential centrifugations at 500× *g*/5 min and 1500× *g*/15 min. Equal volume of 16% PEG, 1M NaCl was added to the post 1500 g supernatant, and the mixture was incubated overnight. The samples were centrifuged at 3500× *g* at 4 °C for 1 h to pellet the EVs as well as cell-free DNA. Pellets were resuspended in 300 µL PBS and were incubated with 0.2% Triton X100 and 10 µg/mL of Proteinase K (Invitrogen, 10665795) for 45 min at 37 °C to digest and dissociate proteins from NAs. An equal volume of Phenol/Chloroform/Isoamyl alcohol (25:24:1), saturated with 100 mM Tris-EDTA to pH 8.0 (Thermo Scientific Chemicals, 327111000), was added to the sample, followed by vortexing and centrifugation at 10,000 g for 5 min. The aqueous SN was transferred to another tube, mixed with an equal volume of chloroform, vortexed and centrifuged again at 10,000× *g* for 5 min. This step was repeated a second time. The upper phase was then transferred to another tube and was supplemented with sodium acetate to a final concentration of 0.3 M and 2.5 vol. of 96% ethanol. The samples were kept at −70 °C for 1 h, centrifuged at 17,000× *g* for 30 min and pellets were dissolved in TE buffer. The samples were directly analyzed for 260 nm absorbance with a NanoDrop™ 2000 instrument and were run on 1% ethidium bromide-stained agarose gel. For DNase/RNase sensitivity analysis, 5 µL of NA sample was supplemented with DNase I buffer (10 mM Tris-HCl (pH 7.5 at 25 °C), 2.5 mM MgCl2, 0.1 mM CaCl2), and either 5 KU DNase I (Sigma Aldrich, D4263) or 10 µg/mL of RNase A (Sigma Aldrich, R-5125). The final volume of the nuclease-treated samples and the controls without nucleases was brought to 10 µL and the samples were incubated for 30 min at 37 °C. Proteinase K (10 µg/mL) was added to each sample to digest and dissociate RNase A from the DNA- and DNase I from the RNA molecules, and the samples were incubated for 15 min at 37 °C. The control and the nuclease-treated samples were subjected to 1% agarose gel electrophoresis, stained with 0.5 µg/mL of ethidium bromide for 15 min and observed on a UV-illuminator.

To detect cell-free nucleic acids associated with large EVs and particles (EVPs), EVPs, isolated with 10,000× *g* centrifugation as described above and resuspended in 1x PBS, were stained with 100 ng/mL of PI (eBioscience™, San Diego, CA, USA, 00-6990-50) and were analyzed with a FACS Calibur instrument (Becton Dickinson, Franklin Lakes, NJ, USA) using low flow rate and a fixed time—50 s. Samples from unconditioned medium were processed in parallel with the EVPs as a control for the presence of contaminating particles in the cell culture medium and the PBS.

### 4.9. Cell Cycle and PI Viability Assay

Cells were treated in 24-well plates (2 × 10^5^ cells/well) in 500 µL of growth medium supplemented with 5% exosome-depleted FBS for 48 h. The cells were washed with PBS, detached with 100 µL of 1X trypsin (Thermo Fisher, 15400054), washed again with PBS, 5% FBS to neutralize the trypsin and resuspended in 400 µL of PBS. The samples were stained with either Propidium Iodide (40 ng/sample) for viability or 62.5 µM DRAQ5 (Invitrogen, 65-0880-96) for cell cycle analysis. The cells were analyzed on BD FACSCalibur™ Flow Cytometer and the data was processed with the Cyflogic software (https://www.cyflogic.com).

### 4.10. Alkaline Comet Assay

The assay was performed according to a previously described procedure [59,60]. The slides were immersed briefly in 1% (w/v normal melting point agarose, their backs were wiped and they were left on a plate overnight at room temperature (RT). The cell suspensions were counted using an automatic cell counter (Countess^®^, Invitrogen) and centrifuged at 300× *g* for 5 min at 4 °C, washed with ice-cold PBS, and centrifuged again. 25 µL of the cell suspension in PBS (~1 × 10^6^ cells/mL) were mixed with 100 µL 0.75% LMP low melting point agarose. Two drops (60 µL) of the mixture were transferred to a pre-coated microscope slide. The gels were covered with coverslips and kept for 5 min at 4 °C in the dark. The coverslips were then removed and the slides were placed in standard lysis solution (2.5 M NaCl, 100 mM EDTA, 10 mM Tris base, pH 10–10.5, with added 1 mL of Triton X-100 and 10 mL DMSO before use) for at least 1 h in a Coplin jar at 4 °C in the dark. The slides were transferred directly to the electrophoresis tank, containing cold (4 °C) 0.3 M NaOH and 1 mM Na2EDTA, pH > 13, and incubated before electrophoresis solution for 20 min. Electrophoresis was performed at ~1 V/cm for 20 min, pH > 13 at 4 °C. The gels were neutralized by washing in cold neutralizing solution (400 mMTris–HCl, pH 7.5) three times for 5 min. Slides were washed for 10 min in cold (4 °C) dH_2_O. The gels were air-dried overnight. The samples were stained with 1X SYBR™ Gold Nucleic Acid Gel Stain (Invitrogen, S11494) for 10 min at RT and visualized on Axiovert 200 fluorescent microscope (Zeiss, Oberkochen, Germany). Images were analyzed using CaspLab—Comet Assay Software (version 1.2.2) [61].

## 5. Conclusions

The current study demonstrates that TMZ and CHQ synergistically activate secretory autophagy in GBM cells, as demonstrated by the enhanced release of EVs containing autophagy markers and cell-free nucleic acids. This effect is not confined to CHQ and TMZ, as shown by treatments with etoposide and Bafilomycin A1, and it is probably driven by DNA damage in the context of lysosomal/late-stage autophagy inhibition. This synergy may be explained by a crosstalk between signaling cascades and mechanisms aimed at alleviation of the cytotoxic effects of the autophagy inhibitors and the TMZ-induced accumulation of damaged DNA/RNA molecules. The synergistic enhancement of EV secretion was also evident in THP-1 monocytes and macrophages treated with TMZ and BafA1, suggesting a broader relevance across cell types. In vivo, both EVs and cell-free NA are known to promote the migratory and metastatic capacity of cancer cells. The significant enrichment of EV-associated caveolin-1, which is associated with cancer aggressiveness and resistance to therapy, further underscores the potential role of these EVs in modulating GBM aggressiveness. Therefore, further research into the mechanisms behind the synergistic intercourse reported here may provide clearer understanding of the overall pharmacological effects of similar drug combinations in GBM and other types of cancer cells.

## Figures and Tables

**Figure 1 ijms-26-09692-f001:**
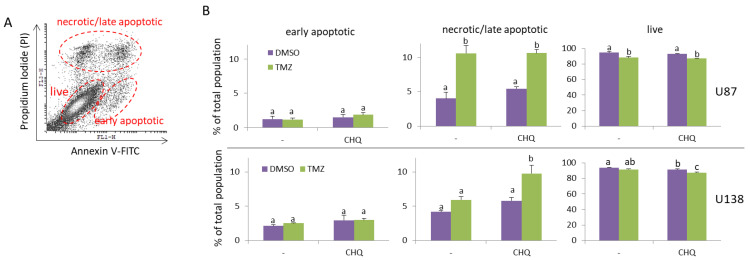
CHQ significantly enhances the cytotoxic effect of TMZ on TMZ-resistant (U138-MG) but not TMZ-sensitive glioma cells (U87-MG). Cells were cultured in medium with 5% exosome-depleted FBS, and were treated with vehicle (0.2% DMSO), 400 µM TMZ, 10 µM CHQ or both, as indicated for 48 h. The cells were stained with Annexin V-FITC and Propidium Iodide (PI) and were analyzed with flow cytometry. (**A**) A representative dot plot illustrating the gating of live, early apoptotic and late apoptotic/secondary necrotic cells. (**B**) The histograms show the average percentages of U87-MG and U138-MG cells within the indicated gates; n = 3; error bars—standard deviation. Data were analyzed with two-way ANNOVA and Tukey’s post hoc test. The means of treatments annotated with the same letter are not statistically significant, *p* < 0.05.

**Figure 2 ijms-26-09692-f002:**
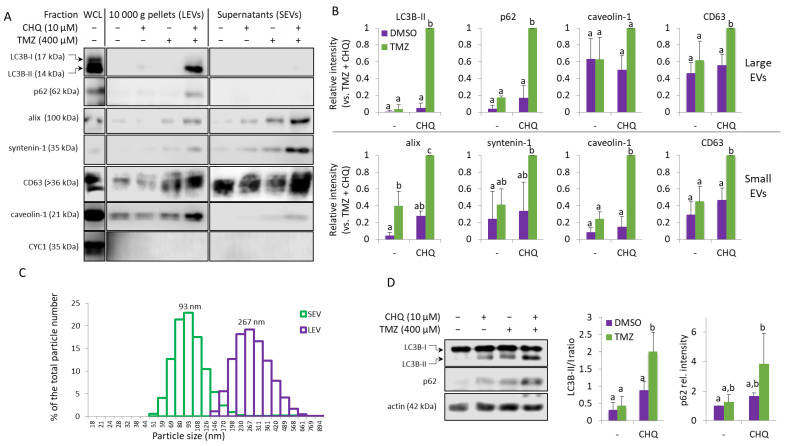
TMZ and CHQ synergistically induce secretion of EVs containing autophagy markers in cultures of U87 MG cells. (**A**) Samples of large (LEVs) and small (SEVs). EVs from cells treated as indicated for 48 h were analyzed with WB, using the indicated antibodies. EVs isolated from equal volumes on conditioned medium were compared in parallel with 10 µg of protein from whole cell lysates (WCL). (**B**) WB densitometry—as some of the markers are undetectable in control samples, the densitometry values are presented relative to the highest intensities—the TMZ + CHQ samples. (**C**) DLS analysis of EVs from the TMZ + CHQ-treated U87 cells. (**D**) WB analysis of LC3B and p62 in WCL of U87 cells treated as in panel (**A**). The histograms to the right show the WB densitometry data. The LC3B-II levels are presented relative to LC3B-I; p62 is normalized to actin and presented relative to its amount in the DMSO controls. For all histograms in the figure, n = 3, error bars—standard deviation; The means of treatments annotated with the same letter are not statistically significant, *p* < 0.05.

**Figure 3 ijms-26-09692-f003:**
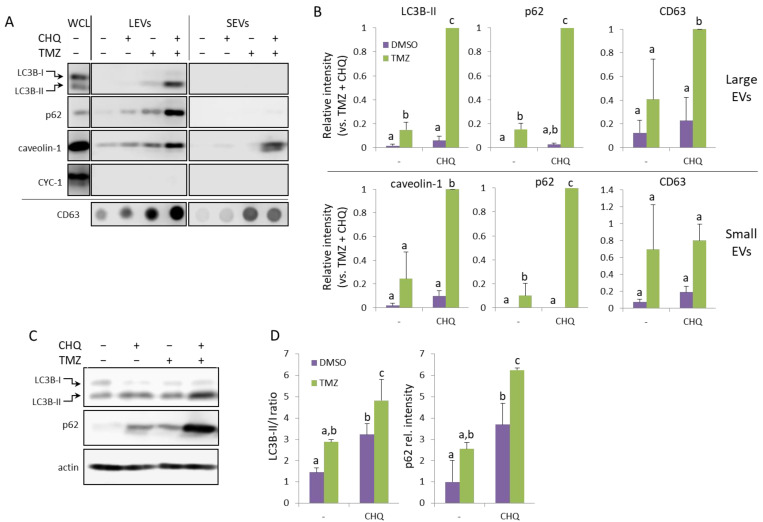
TMZ and CHQ synergistically induce secretion of EVs containing autophagy markers and caveolin-1 in U138 cell cultures. Panels (**A**,**B**)—representative Western and dot blot images and densitometry data, respectively, of EV samples. Panels (**C**,**D**)—WB images and densitometry data, respectively, of WCL from U138 cells. The samples were obtained and processed as described in Figure 2 for U87-MG cells; n = 3, error bars—standard deviation; The means of treatments annotated with the same letter are not statistically significant, *p* < 0.05.

**Figure 4 ijms-26-09692-f004:**
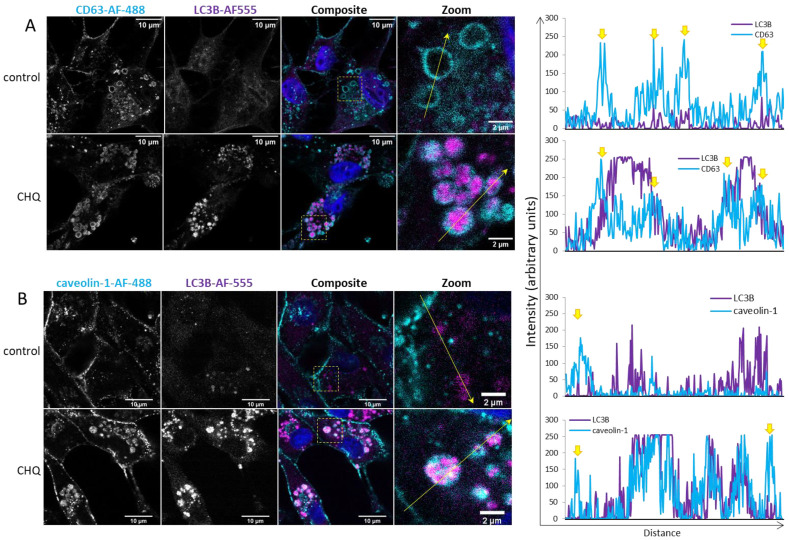
CHQ induces accumulation of LC3B in the lumen of CD63-positive amphisomes/autolysosomes; LC3B and caveolin-1 colocalize within the CHQ-induced LC3B aggregates. (**A**) Controls, and cells treated with 10 µM CHQ, were fixed 48 h post treatment. CD63 was stained with mouse anti-CD63 and secondary Alexa Fluor 488 antibodies (cyan); LC3B was labeled with rabbit anti-LC3B and secondary Alexa Fluor-555 antibodies (magenta); nuclei were stained with DAPI (blue in the composite image). The images were obtained with a Leica Stellaris 8 confocal system. The positions of the zoom areas are indicated with dashed rectangles in the composite images. Arrows indicate the positions and the directions of the intensity profile lines plotted to the right of the images. The yellow arrows above the CD63 intensity profiles indicate the limiting membranes of the CD63-positive vesicles. (**B**) U87 cells were treated and analyzed as in panel (**A**) using rabbit anti-caveolin-1 and mouse anti-LC3B antibodies. The yellow arrows above the caveolin-1 intensity profiles indicate the plasma membrane.

**Figure 5 ijms-26-09692-f005:**
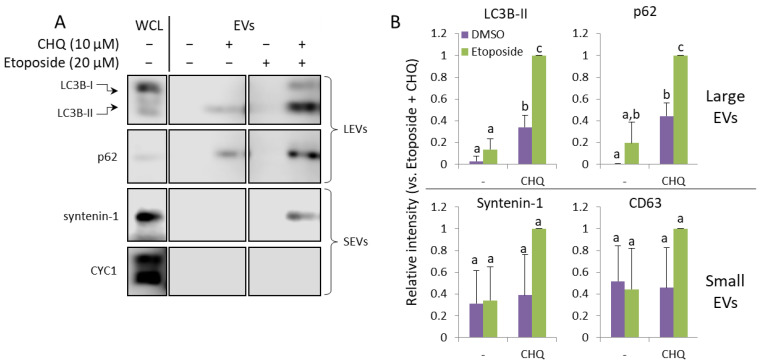
CHQ and etoposide synergistically upregulate LC3B-II and p62 in LEVs. (**A**) WB analysis of EV samples from U87 cells treated as indicated for 48 h. (**B**) Densitometry analysis of the immunoblots; n = 3, error bars—standard deviation; The means of treatments annotated with the same letter are not statistically significant, *p* < 0.05.

**Figure 6 ijms-26-09692-f006:**
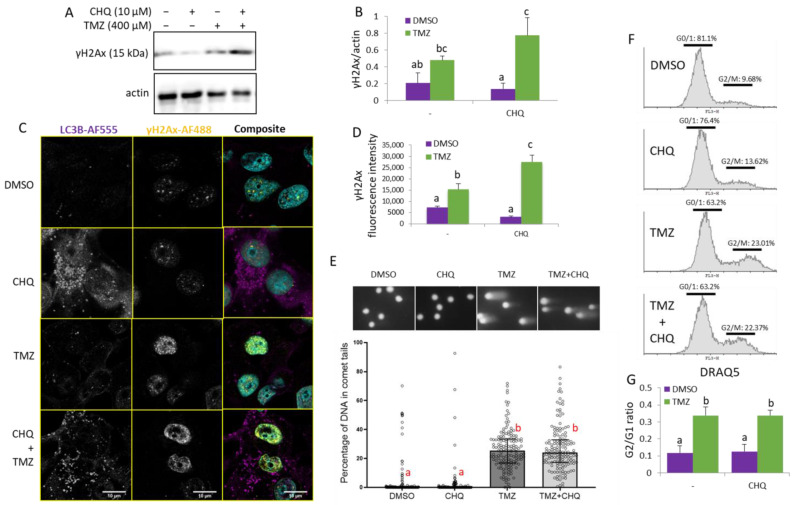
CHQ enhances the TMZ-induced DDR as indicated by γH2Ax upregulation, but it does not significantly affect the accumulation of damaged DNA in the cells and the G2/M cell cycle arrest of U87 cells. (**A**) The γH2Ax accumulation was analyzed with WB in cells treated as indicated; actin was used as a loading control. (**B**) WB densitometry of the γH2Ax signal, normalized against actin, n = 3, error bars—standard deviation. (**C**) Confocal imaging of U87 cells, treated as indicated, and immunostained with antibodies against LC3B and γH2AX (rabbit and mouse, respectively). The samples were stained with secondary anti-rabbit AF-555 and anti-mouse AF-488-conjugated antibodies (purple and yellow pseudo-colors, respectively); DNA was stained with DAPI (cyan). (**D**) Quantification of the γH2Ax fluorescence intensity values in cell nuclei; the histogram shows the average γH2Ax fluorescence intensity per nucleus; n = 10; error bars—standard deviation. (**E**) U87 cells were analyzed with a comet assay, and the percentage of DNA in comet tails was calculated with the CaspLab software. The bars in the scatter plot below the representative images show the median values; the error bars indicate the quartile range, n = 150. (**F**) Cell cycle distribution of cells, treated as indicated in the histograms, was analyzed with DRAQ5 staining and flow cytometry. (**G**) The bar graph shows the ratio of cells in G2/M vs. G0/1 measured as indicated in panel (**F**), n = 3, error bars, standard deviation. In all of the histograms in this figure, the means of treatments annotated with the same letter are not statistically significant, *p* < 0.05.

**Figure 7 ijms-26-09692-f007:**
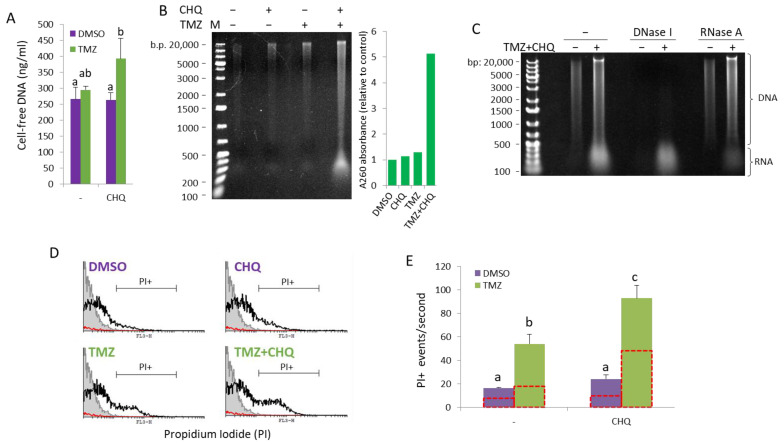
CHQ and TMZ synergistically induce secretion of cell-free nucleic acids in U87 cultures. (**A**) Cell-free DNA was directly measured in conditioned supernatants of cells grown in FluoroBrite™ DMEM using a Qubit™ 1X dsDNA High Sensitivity (HS) assay kit. (**B**) Cell-free NA were extracted from conditioned medium using phenol–chloroform–isoamyl alcohol, and were visualized on an EtBr-stained agarose gel. The histogram to the right of the gel image shows the A260 absorbance values as an indicator for the total NA amount relative to DMSO control. (**C**) Cell-free NA from control and TMZ + CHQ cells were treated with DNase I or RNase A, as indicated, prior to gel electrophoresis, demonstrating the presence of both DNA and RNA molecules in the TMZ + CHQ-treated samples. (**D**) Detection of cell-free nucleic acids associated with large EVs and particles (EVPs) through flow cytometry. EVPs, isolated with 10,000× *g* centrifugation, were stained with PI and were analyzed with a FACS Calibur instrument using low flow rate and a fixed time. Samples from unconditioned medium were processed in parallel with the EVPs as a control for presence of contaminating particles in the cell culture medium and the PBS. The representative overlay histograms show the non-stained control (filled gray contour), stained EVP samples—open black contour and the PBS control—red contour. (**E**) The bar graph shows the average PI+ events/second, gated as shown in panel (**D**). The open red contour bars indicate the number of Triton-resistant PI+ particles, following treatment of the samples with 1% Triton X100. n = 3; error bars—standard deviation. In all of the histograms in this figure, the means of treatments annotated with the same letter are not statistically significant, *p* < 0.05.

**Figure 8 ijms-26-09692-f008:**
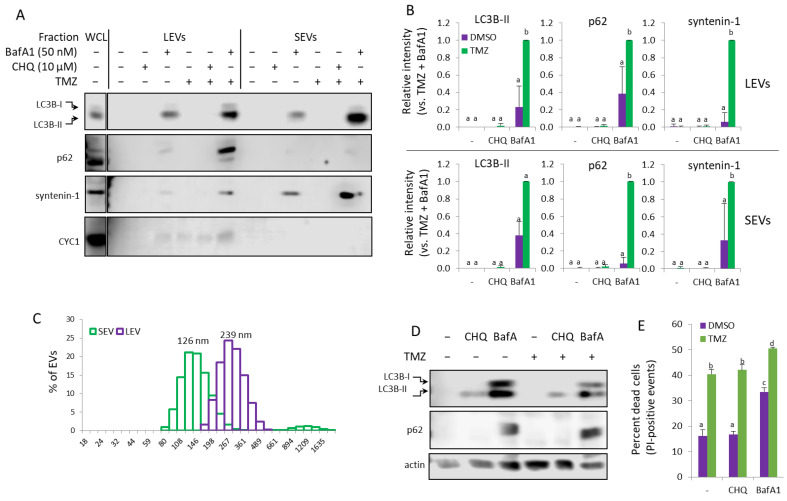
In THP-1 monocytes, Bafilomycin A1 but not CHQ synergizes with TMZ to induce secretion of EVs. (**A**) EVs from THP-1 cells, treated as indicated, were analyzed with WB using the indicated marker antibodies. (**B**) Densitometry of the markers detected in the EV samples. (**C**) DLS analysis of EVs from THP-1 cells, treated with TMZ + BafA1. The samples show typical size distribution profiles for SEVs and LEVs, with modes of 126 and 239 nm, respectively. (**D**) Representative WB images showing LC3B and p62 in WCL of THP-1 cells, treated as in panel (**A**). (**E**) Cytotoxicity assay using PI staining and flow cytometry. The bar graph shows the percentage of dead (PI-positive) cells. For all histograms in the figure, n = 3, error bars—standard deviation; The means of treatments annotated with the same letter are not statistically significant, *p* < 0.05.

**Figure 9 ijms-26-09692-f009:**
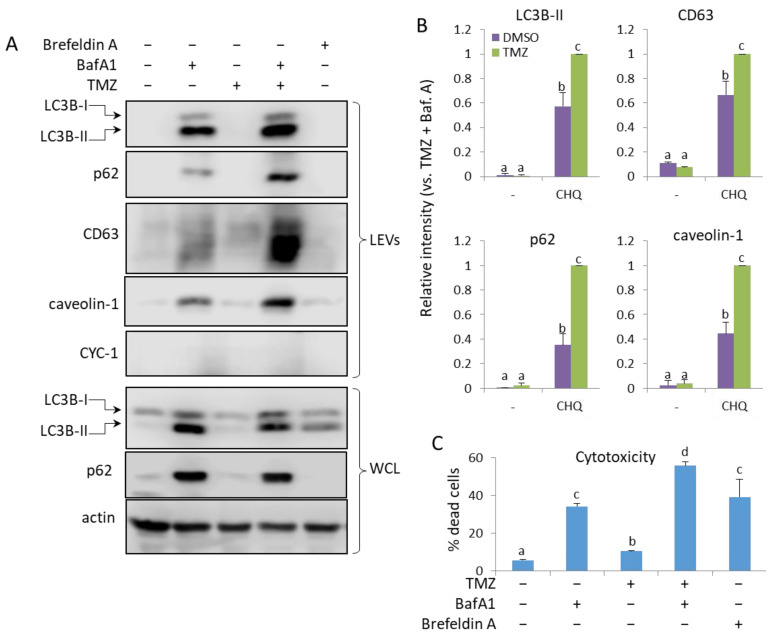
TMZ enhances the BafA1-induced secretion of LC3B-II, p62, CD63 and caveolin-1 in LEVs from U87 cells. (**A**) WB analysis of LEVs and WCLs from cells, treated with 50 nM BafA1, 400 µM TMZ and 3 µg/mL of Brefeldin A, as indicated. (**B**) WB densitometry analysis showing the relative intensity of the detected signals in LEVs. (**C**) Cytotoxicity of the treatments measured with PI staining and flow cytometry. The bar graph shows the percentage of dead (PI-positive) cells, n = 3, error bars—standard deviation. The means of treatments annotated with the same letter are not statistically significant, *p* < 0.05.

## Data Availability

The original contributions presented in this study are included in the article/Appendix A. Further inquiries can be directed to the corresponding author.

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
