# Peer review of "Synergistic Upregulation of Extracellular Vesicles and Cell-Free Nucleic Acids by Chloroquine and Temozolomide in Glioma Cell Cultures"

_ijms, 2025, doi:10.3390/ijms26199692_

Round 1

Reviewer 1 Report

Comments and Suggestions for Authors

The authors describe a synergistic relation between DNA damage induction and lysosomal inhibition in inducing secretory autophagy and EV release in GBM cells. Using in vitro assays to assess apoptosis, DNA damage, nucleic acid release, and by using quantitative and qualitative protein estimation by immunoblots and ICC, the authors first determine that Chloroquine (CHQ) and Temozolomide (TMZ) treatment together induce EV secretion in U87/U138 GBM cells. They go on to show that even though CHQ does not enhance TMZ-induced DNA damage or G2/M arrest, the CHQ+TMZ-induced Calveolin+/LC3B-II+ EVs contain significant upregulation of cell-free nucleic acids, relative to individual treatments. This EV release effect is not specific to TMZ, but also occurs upon co-treatment of CHQ with Etoposide, an alternative DNA-damage inducing agent. EV induction was also observed in THP-1 monocytes upon co-treatment with TMZ and BafilomycinA, but not with low dose CHQ, indicative of cell type dependency.

Presented only as an in vitro report, the study is relatively well designed. The manuscript sheds light on a potential novel mechanism that GBM cells might use in response to TMZ/CHQ treatment, with implications to treatment resistance.

The study is of broad interest to the field, but it’s impact on glioma research is limited due to the reasons addressed below:

-The study shows data from 2 commercially available GBM cell lines. It is suggested that the authors add 1-2 patient-derived cell lines, to increase the robustness of their conclusions. Inclusion of patient-derived cells will make the data stronger, and might be able to delineate differences in EV induction responses (eg. Differences in autophagy dependency).

- Fig 1- How were the TMZ/CHQ doses determined? Are the ic50 doses for TMZ and CHQ different in U87 and U138 cells? Please include this information (for all drugs used) as supplemental data.

-Figure 1b- Please provide actual percentage changes in apoptotic and live cells in the text. The smaller changes cannot be discerned clearly from the graphs provided. The difference between live cells in TMZ treated (+/-CHQ) in U138 might be statistically significant as described, but is very minimal to discern if it might have any biological significance.

-What was the loading/normalization control used for the SEV/LEV immunoblots (total protein stain, etc)? Please include details of quantification in the methods section.

-Figure 3A- Please include CD63 in the immunoblot, similar to that shown for U87 cells

-Figure 6C, D- was yH2Ax intensity measured across the field or per cell?

-Please introduce the rationale for brefeldin use in the manuscript text (section 2.8), before it is used in figure 9, for readers who might not be well versed in its use.

-TMZ efflux through EVs has been implicated in GBM treatment resistance. The current study can be made stronger by measuring TMZ concentrations in the cells/EVs in the different conditions to determine whether TMZ/CHQ-induced secretory autophagy is a mechanism of drug resistance in these cells.

-The manuscript would be strengthened by inclusion of an in vivo intracranial model to determine (a) whether there is enhanced EV secretion in the plasma/CSF/GBM microenvironment after TMZ/CHQ co-treatment, and (b) how it relates to disease progression and overall survival in the model. It is suggested that the authors include an essential in vivo experiment to substantiate the conclusions form the in vitro data.

Reviewer 2 Report

Comments and Suggestions for Authors

Review Comments: [IJMS] MDPI

The authors presented a topic titled “Synergistic upregulation of extracellular vesicles and cell-free nucleic acids by chloroquine and temozolomide in glioma cell cultures”. Overall, the research article aims to reveal the interplay between the combination of chloroquine and temozolomide, and the activation of unconventional secretory pathways along with the extracellular vesicles upregulation in glioblastoma cells. Although the article topic is relevant for molecular sciences research community and the IJMS journal scope, there are a few minor points that require careful consideration. These points are including the following:

  1. Introduction section:
    1. The authors should further elaborate and provide more details on the relationship between GBM and autophagy with respect to the TMZ and CHQ, in the paragraph beginning line 63.
    2. In line 64, The authors should cite more than one reference to provide stronger evidence as citing only one study may not adequately support the statement presented. "Studies have implicated autophagy in the etiology of GBM and the development of resistances to treatment [13]" Please ensure these are included to reflect the state of the art in the molecular sciences research area. The main aim of IJMS journal is to provide experimental and theoretical highlights in as much detail as possible.
    3. In line 41, reference [2] would be more appropriately placed at the end of the sentences in line 42 for clarity and consistency.
    4. In line 64-66, Please consider moving references [12] and [14] to the end of the sentences in line 66 for consistency.
  2. Discussion section:
    1. In line 340, authors should remove the comma before ‘as well as,’ as it is unnecessary.
    2. In line 390, Please consider moving reference [41] to the end of the sentence in line 391 “Moreover, EVs, including exosomes, are known 390 to serve as vehicles for export of damaged DNA [41], as well as RNA:DNA hybrids [42]”.
  3. Conclusions section:
    1. The authors should provide more details in the Conclusions section to strengthen the summary of their research study and highlight the key findings and their implications.
    2. In line 584, authors should add comma after caveolin-1 in this sentence “autophagy markers, caveolin-1 and cell-free nucleic acids.”
  4. Language and punctuation: Please consider revising the research article for clarity and punctuation, particularly commas.
  5. References sections: References should be recent and updated to cope with the IJMS Journal.
